# Healthcare University Courses Fail to Improve Opinions, Knowledge, and Attitudes toward Vaccines among Healthcare Students: A Southern Italy Cross-Sectional Study

**DOI:** 10.3390/ijerph20010533

**Published:** 2022-12-28

**Authors:** Marco Palella, Chiara Copat, Alfina Grasso, Antonio Cristaldi, Gea Oliveri Conti, Margherita Ferrante, Maria Fiore

**Affiliations:** 1Medical Specialization School in Hygiene and Preventive Medicine, Department of Medical, Surgical Sciences and Advanced Technologies “G.F. Ingrassia”, University of Catania, Via Santa Sofia 87, 95123 Catania, Italy; 2Department of Medical, Surgical and Advanced Technologies “G.F. Ingrassia”, University of Catania, Via Santa Sofia 87, 95123 Catania, Italy

**Keywords:** vaccine hesitancy, healthcare students, vaccination, public health

## Abstract

(1) Background: Healthcare providers have a crucial role in contrasting vaccine hesitancy (VH). We aimed to investigate opinions, knowledge, and attitudes toward vaccines in healthcare students (HS) at the University of Catania (Italy). (2) Methods: A survey was conducted from 1 October 2019 to 31 January 2020. Data on the opinions, knowledge, and attitudes of HS toward vaccinations were collected using an anonymous self-administered questionnaire. The opinion answers were added to calculate the VH index (<18 = low, 19–22 = medium, >23 = high). Data were summarized by the VH index, degree, year of study, and sex differences, using descriptive statistics. (3) Results: A total of 1275 students (53.7% females) participated in the study, with a median (IQR) age of 21 (19–22) years. The median level of VH was 20 (17–23), with slightly higher values in males. We found an inverse trend between VH and opinions, knowledge, and attitudes toward vaccines. The same trend was confirmed in all study courses. Furthermore, the comparison between sexes revealed a higher level of knowledge in women. (4) Conclusions: The results highlight a lack of knowledge about vaccines, as well as contrasting opinions and attitudes among future health professionals. Therefore, future interventions on these topics in the preparation of future healthcare providers are needed.

## 1. Introduction

Vaccination is one of the most important instruments to control infectious diseases [1] and a relevant achievement of public health [2]. Although vaccines are estimated to avoid between 2–3 million deaths each year [3], people who are hesitant to accept vaccination choose to delay or refuse vaccines for different reasons, including political, scientific, and religious ones [2,4]. Vaccine hesitancy (VH) is a global, complex, and constantly changing phenomenon, currently representing one of the most significant public health problems. In 2019, the World Health Organization (WHO) classified VH in its top ten threats to global health [5] and defined it as the ‘‘delay in acceptance or refusal of vaccines despite availability of vaccination services” [6].

The WHO has provided many recommendations to understand VH, its determinants, and challenges to contrast it. Recognizing the importance of this phenomenon in achieving the established health goals, the WHO Strategic Advisory Group of Experts (SAGE) on immunization emphasizes the urgent need to develop institutional systems and organizational capabilities at the local, national, and global levels to proactively identify, monitor, and address VH in order to respond promptly to anti-vaccination movements in the event of misinformation or potential adverse events following vaccination [7].

Moreover, the SAGE investigates the nature and extent of VH and has developed a model of the determinants of VH that focuses on three domains. 1. Contextual influences; 2. Individual/social group influences; and 3. Vaccine and vaccination-specific issues. All three domains include the influence of others on vaccine hesitancy. Domain 1 includes influential leaders and individuals; Domain 2 includes personal experiences with and trust in the healthcare system and providers; and Domain 3 includes the role of healthcare professionals [4].

Many factors may play a key role in the apparent increase in this phenomenon in the world, and nowadays the media and communication of fake news play an important role [8]. Moreover, the importance of healthcare providers (HCPs) regarding vaccine recommendations in the decision-making process has been well documented [9]. For example, pregnant women have been shown to take up pertussis vaccination after HCPs counseling and their willingness to vaccinate during pregnancy or to take persuasive action to reduce vaccine hesitancy increased [10]. Lin et al. [11] reviewed some factors related to HCPs that may lead to vaccine hesitancy. For example, a recommendation was positively associated with provider knowledge and experience, beliefs/opinions about disease risk, and perceptions of vaccine safety, necessity, and efficacy [12,13]. HCPs’ vaccination attitudes and practices differed by specialties, vaccines, and countries, whereas demographic effects were inconclusive [14,15,16]. Barriers include anticipation of patient/parental concerns or refusal, a lack of clear guidelines, a lack of time, and cost, which play a role in hesitancy. In addition, studies indicated that provider hesitancy was related to inadequate knowledge, low confidence in the vaccine, and suboptimal uptake [17,18,19,20]. The growing interest in vaccine hesitancy has led to the development of different tools and strategies to encourage vaccination acceptance. In order to improve knowledge about and attitudes toward VH, many HCPs have proposed different ways to contrast vaccine hesitancy, such as transparency in policy decisions, education, and information to the public and healthcare providers about this process [21]. Schools and universities play a fundamental key role in this process [22,23]. In particular, Costantino et al. [22] and Underwood et al. [23] highlighted the importance of a training period for student’s to improve their knowledge about vaccines, in order to reduce vaccine hesitancy and to contrast beliefs created by fake news. Especially for health sector students, and in general for HCPs, the WHO confirms the role of education as an essential component of all strategies to improve health professional skills [3,5].

First and foremost, communication is one of the necessary elements that should be trained to increase knowledge and empowerment, promote health-promoting attitudes, encourage changes in social norms, and facilitate access to and ensure adherence to prevention and treatment programs.

Second, the process of vaccine hesitancy includes different individual or group factors related to a vaccine or vaccination in general. Any intervention must address the specific factors identified for VH and may be necessary not only at the individual level (dialogue and better information), but also at the population level. It is important for HCPs to know details about perceptions, attitudes, knowledge, and behaviors to counteract procrastination. Nowadays, it is also useful to consider the role of social media and community sites in encouraging VH to counter false beliefs, new determinants of vaccination refusal, false myths, and misinformation. It is important to underline that the key is not to criticize hesitant individual. In order to reduce their VH, we should answer their questions and provide clear and simple information for an informed and conscious decision [3,5]. This phenomenon emerged in the last years and involves HCPs. For this reason, it is useful to evaluate the vaccination opinions, knowledge, and attitudes of young future HCPs. In fact, it has been demonstrated that young people were more likely to overcome their vaccine hesitancy than adults [24].

Moreover, in the last three years, the COVID-19 pandemic has strengthened vaccine hesitancy involving almost all population categories including also students and doctors, making it clear how the importance of correct knowledge is essential to counteract this current phenomenon [25,26,27]. The aim of this study was to investigate opinions, knowledge, and attitudes toward vaccines among healthcare students at the University of Catania.

## 2. Materials and Methods

### 2.1. Study Design, Setting, and Participants

We conducted a cross-sectional study from 1 October 2019 to 31 January 2020. The reference population investigated in this study consisted of students in the health profession courses (degrees in physiotherapy, nursing, obstetrics, motor sciences, and biomedical laboratory techniques), and students in medicine and surgery at the University of Catania. Participation in the study was proposed to all students involving all the first-, third-, and fifth-year medical students and the first-, second-, and third-year of the other degree. The involvement in the study was voluntary, anonymous, and without risks for the participants. The non-adherents were only those students who were absent at the time of recruitment.

Participants filled in the questionnaire during classroom lecture time after the aims of the project have been explained. All procedures were performed according to the ethical standards of the 1975 Helsinki Declaration, as revised in 2013. Ethical approval was given by the Ethics Committee of the University Hospital “Policlinico G. Rodolico—S. Marco” of Catania. Written informed consent was obtained from all participants.

### 2.2. Measurements

All recruited students were asked to fill in an anonymous questionnaire developed based on the questionnaire validated by the National Institute of Health for the European project “HproImmune—Promotion of the immunization of health workers in Europe” (HproImmune, 2020; Epicenter, 2014). The validation of the questionnaire was carried out by verifying the understanding of the questions on a small pilot sample of future health professionals using the pre-test method.

The questionnaire, which was designed to collect opinions, knowledge, and attitudes of health professionals toward vaccinations, also included the collection of socio-demographic data (gender, age, degree, and year of study) (Appendix A). The questionnaire was completed in paper form and then transferred to an electronic Excel database.

### 2.3. Statistical Analysis

Statistical analysis was performed using SPSS 27.0 (SPSS Inc., Chicago, IL, USA). We summarized quantitative variables by median (IQR), whereas for qualitative variables, we reported the absolute and relative frequencies. The results of the descriptive analysis were summarized by vaccine hesitancy, degree, and year of study, focusing on sex differences. The answers related to the question “Give your opinion on the following statements: …” (Appendix A), expressed on a Likert scale (from 1 “Strongly disagree” to 5 “Strongly agree), have been added together so the resulting variable has been categorized into tertiles, which in turn, was used to calculate the “vaccine hesitancy index” that increases with VH (<18 = low VH, 19–22 = medium VH and >23 = high VH). In order to evaluate the internal consistency and reliability of each domain of the questionnaire (opinions, knowledge, and attitudes), we used Cronbach’s alpha coefficient. High values have been considered as those above 0.70. Then, through the bivariate analysis (the chi-square test and the Kruskal–Wallis test), the association between the vaccine hesitancy categories (low, medium, and high) and the variables investigated by this study (opinions, knowledge about the vaccines recommended for healthcare professionals, and their attitudes toward vaccinations) were studied. Cramer’s V was used to quantify the strength of the relationships between the qualitative study variables.

## 3. Results

A total of 1275 students (53.7% females) completed the questionnaire, with a response rate of 75%. Socio-demographic characteristics of the responders are shown in Table 1 and are stratified by vaccine hesitancy level.

The median (IQR) level of VH was 20 (17–23) with slightly higher values in males (median: 20, IQR 18–24).

Table 1 shows higher levels of VH among younger students, as well as among undergraduate students in nursing, obstetrics, physical education, and biomedical laboratory technicians. No association was found between the VH index and sex (*p* = 0.86), whereas an association was highlighted with the degree (*p* = 0.00; Cramer’s V 0.24), even though it explains only 6% of the variance. It is interesting to note that the highest frequency of subjects with a “high” level of VH is found in the medicine, nursing, and physical education degrees (Table 1).

### 3.1. Healthcare Students’ Opinions

The opinion level internal consistency analysis (11 Likert scale variables) (Appendix A) showed a standardized Cronbach’s alpha equal to 0.70, corresponding to high reliability. Appendix A shows a weak negative relationship between VH and the percentage of students who believed in the importance of vaccines (100% vs. 99.8 vs. 98.8%) and their usefulness (99.1% vs. 99.5% vs. 98.8%), as well as in their effectiveness (100% vs. 100% vs. 99.5%), whereas a positive relationship was found between VH and the percentage of “agree + completely agree” responses to the statements “I believe more in natural immunity acquired through disease than in vaccines” (0% vs. 2.3% vs. 11.2%), “I’m afraid of the side effects” (0% vs. 4.1% vs. 21.2%), “I don’t think I’m at risk of contracting any infectious disease” (0.5% vs. 6.4% vs. 11.5%),”I’m afraid of getting sick after getting vaccinated” (0% vs. 0% vs. 11.2%), and “I am wary of the long-term health effects of vaccinations” (0.2% vs. 0.9% vs. 7.2%) (Appendix A). Most of the students (75.8%) believed that the vaccination of healthcare professionals is a prerequisite for working in the health sector. In particular, as VH increased, the percentage (99.5% vs. 99.5% vs. 94.3%) of respondents who considered vaccination a prerequisite for working in the health sector decreased (Figure 1a) (Appendix A). Slightly more than half (56.0%) of the respondents believed that vaccination is a duty for health professionals, as they should be an example to patients. However, as VH increased, the percentage (97.5% vs. 95.7% vs. 86.3%) of students who considered vaccination as a duty of health professionals decreased (Figure 1b) (Appendix A). The answers relating to the question “I believe health worker vaccination are…“A prerequisite for working in the health sector or a duty of healthcare professionals as they should be a role model for patients” showed the same trend in all studied programs (data not furnished). Most of the students (88.4%; males 84.9% vs. females 91.4%) answered “yes” to the question “In your future clinical practice, would you recommend vaccinations to your patients?”.

We also found that the percentage of students who would recommend vaccination decreased significantly with increasing VH (97.7% vs. 92.2% vs. 74.1%; *p*-value <0.001, Cramer’s V 0.23). It is interesting to note that with increasing VH, the number of students who responded to the question “It is not within my remit” increased significantly (2.3% vs. 5.9% vs. 13.7% *p*-value < 0.001, Cramer’s V 0.23), with the exception of medical and obstetrics students. Despite a low VH (VH <18), almost half (40%, *p*-value 0.07, Cramer’s V 0.43) of the students in a degree in laboratory techniques felt that vaccination was not their responsibility.

In particular, students in the medical (97.1%), obstetrics (92.7%), nursing (85.7%), and physiotherapy (84.5%) courses answered “Yes” more frequently than the other courses, whose frequencies exceeded 50% in each case. Appendix A show the distribution of the vaccination refusal index for each degree by the willingness to recommend vaccination in their future clinical practice. Overall, students with a medium to high VH level answered “sometimes”, “it is not within my competence”, and “I’m not sure”; conversely, students who answered “yes” had a slightly lower VH level.

From the first to the third year of medical school, no differences were found in the willingness to recommend vaccination in their future clinical practice (96.1% vs. 96.8%), whereas it increased slightly in the fifth year (98.5%).

In Appendix A, we have presented the medical students’ VH distribution by year of study; unfortunately, the comparison between the first year and the subsequent third and fifth years did not reveal any significant reduction in the VH level (Appendix A). Moreover, from the first to the third year, there seems to be a decrease in VH in the group of students who answered “sometimes,” but in the same group, the level of VH increased in the fifth year.

### 3.2. Healthcare Students’ Knowledge

Knowledge levels (nine Likert scale variables) and an internal consistency analysis (Appendix A) yielded a standardized Cronbach’s alpha of 0.80, which corresponds to high reliability.

Table 2 shows that the frequency of students who answered “yes” to the question “Do you know which of the following vaccinations are recommended for healthcare professionals?” decreased as VH increased, with the exception of Hep A vaccination. Conversely, the frequency of students answering “No” increased with VH for some vaccines (influenza, chickenpox, MMR, and meningococcal), whereas the opposite was true for some vaccines (Hep A, Tdap, and pneumococcal), or there was no difference (Hep B). The frequency of students who answered “no” increases together with VH. Different levels of knowledge were found between sex for all vaccines studied (Table 3).

Analyzing the knowledge after stratification by degree, the inverse trend with respect to VH is not always confirmed (Appendix A).

We found an increase in “yes” responses for medicine course years related to influenza (first year 52.1% vs. third year 63.1% vs. fifth year 63.7%) and Hep B vaccinations (first year 78.4% vs. third year 89.8% vs. fifth year 96.0%). Surprisingly, there are still students in the fifth year who answered “no” or “I don’t know” when asked about the flu vaccination recommendation (no 14.9% and I don’t know 21.4%).

### 3.3. Healthcare Students’ Attitudes

The attitude levels (10 Likert scale variables) analysis of internal consistency (Appendix A) showed a standardized Cronbach’s alpha equal to 0.78, corresponding to high reliability.

A total of 66 students, equal to 5% of our sample, were not vaccinated with flu vaccine, while 85.4%, 71.8%, 76.3%, and 51.3% were vaccinated against MMR, HepB, Tdap, and chickenpox, respectively. For the flu vaccine, no overall trend was recorded with respect to the VH, except for vaccinated students in the last year (VH < 18: 5.1%, VH 19–22: 5.7%, VH > 23: 2.7%).

Two hundred and fifty-six (20.1%) students, in particular 34.7% from the medicine degree, could not recall having been vaccinated against Hep B. It is interesting to note at the same time that an important percentage, equal to 13.1%, reported not knowing that they had been vaccinated against Tdap.

We did not analyze the relationship between the student’s VH and the other vaccinations investigated because they are provided to all newborns.

Subjects who reported not being vaccinated in response to the question “If you answered no to the previous question, are you going to get vaccinated in the next few months?” answered “no” for the flu (46.2%), chicken pox (84%), and MMR (50%), whereas 31.8% for Hep B and 36.7% for Tdap replied “I don’t know”. Only 9.2% answered “yes” for the influenza vaccine, 20.8% for MMR, 55.6 for Hep B, and 32.7 for Tdap. No differences were found between males and females.

### 3.4. Healthcare Students’ Possible Reasons for Not Becoming Vaccinated

Regarding the question “If you have not received any of the vaccines mentioned in question 5 please indicate the possible reasons listed”, the most common reason given by students regarding flu vaccination was “I have not had time” (28.2%), with a percentage that increases with increasing VH, or “The vaccine was not free” (75.0%). The answers “I did not know where to go to get vaccinated” and the “procedure is too complex” related to the flu vaccine were not evaluated due to very low frequency. For the same reason, data on the other vaccines studied are not provided.

### 3.5. Association between Specific and Potential Adverse Effects Perception

In response to the question “How do you perceive the association of specific and potential adverse effects” most students answered “not very probable” for “Flu vaccine and Guillain-Barré syndrome (1096, 86.0%)”, “Hep B and Multiple Sclerosis (1093, 85.8%)”, “adjuvant and narcolepsy (963, 75.6%)”, “HPV and Multiple Sclerosis (1074, 84.3%)”, and “vaccine adjuvant and long term complication (875, 68.7%)”, respectively. We found that 28.7% (366) of students answered “medium probable” to the option “vaccine adjuvant and long term complication”.

With respect to VH, we found an inverse trend for the “not very probable” response and an equal trend for the “medium probable” response for all potential adverse effects examined (Appendix A). For the “highly probable” response, we found an inverse trend for all potential adverse effects studied, except for “influenza vaccine and GB syndrome.”

## 4. Discussion

The objective of this study was to investigate opinions, knowledge, and attitudes toward vaccines among healthcare students at the University of Catania.

The results showed that students had an intermediate VH level and that there was a relationship between health professions students’ opinions, knowledge, attitudes, and VH level, especially among the youngest students in medical, nursing, and physical therapy programs. In addition, women seemed to have more confident opinions than men, and their knowledge was more often higher than men’s.

### 4.1. Opinions

Our results highlighted that a part of the students did not consider the vaccination of health professionals as a prerequisite for working in the healthcare sector. It is known from the literature that there may be several factors associated with non-positive opinions among young people and that they are generally context-specific, and vary according to time, place, and vaccine type [28].

In addition, we must keep in mind that our sample also consisted of first-year students, i.e., adolescents (19 years old). Compared with other age groups, there are few studies on adolescent/student vaccination hesitancy and its determinants [29,30]. La Fauci et al. [29] found in healthcare professional students a general lack of confidence and insecurity. In the same way, [30] suggested that future HCPs are not well-prepared to face vaccine-related topics.

Undoubtedly, the most important determinants of the VH of adolescents/young students to vaccinate are a lack of confidence and information problems, as demonstrated in many studies [31]. Karafillakis et al. [31] suggest that information issues are the most important determinants of VH among adolescents followed by concerns about vaccine safety and logistical barriers.

Literature shows that slightly more than 50% of students consider the vaccination of health professionals as their duty, as they should be role models for patients [29]. Unfortunately, about 40% of students participating in our survey did not seem to have understood the importance of the role of HCPs, who, as the literature shows, are the most important and reliable source of information on protection against vaccine-preventable diseases [31,32,33].

### 4.2. Knowledge

We investigated students’ knowledge of vaccinations recommended by health professionals based on Italian Legislative Decree 81/2008, and our results showed an inverse trend in VH and correct answers.

According to different studies [29,30,34], most of the students gave the correct answers, especially for the Hep B and Tdap vaccines, as confirmed by the literature, but the knowledge was not correct for the other vaccines, especially for the flu vaccine [29].

It has been suggested that medical students are better prepared than other majors because of the studies conducted during their university careers, as noted by Dybsand et al. [30].

According to La Fauci et al., it is interesting to note that females had a deeper knowledge than males in relation to all questions included in our survey [29]. This is important to the vaccination process because it has been shown that HCPs who know more about vaccines are also more likely to recommend vaccination to their patients [11]. In addition, the likelihood of being vaccinated appears to be significantly higher if the doctor begins the consultation with a confidential approach about vaccination, although more support is needed, especially for conducting difficult conversations with a patient or a parent who is unwilling to be vaccinated [35]. Knowledge also means clear guidelines for recommended vaccines as key to combating VH.

### 4.3. Attitudes

We found that students had poor attitudes toward the flu vaccination, both in terms of accepting and recommending it in their future clinical practice. This attitude did not seem to improve during the course of the study.

As described in the literature [29,34], many students found the effectiveness of the influenza vaccine limited. In other studies, there are no high percentages of compliance with influenza vaccination, as in Austria, where it is 8–9% [36], whereas in another Italian study, the percentage of vaccination coverage is about 15% and the willingness to accept vaccination for a long period of time seems to be low, as it is only 4% [29].

It is surprising that a proportion of medical students did not know that they have been vaccinated against Hep B, although vaccination for newborns has only been mandatory since 1991. Conversely, other studies have found a low coverage rate for Hep B, but in this case, a greater percentage of vaccine refusers would vaccinate in the following months [35].

We found that females recommended vaccinations more than men, as was found in other studies, although in some of them it seems to apply only to some vaccinations. It is necessary to point out that there is no difference between the sexes in terms of flu vaccinations [29].

It is likely that the low willingness to provide and recommend influenza vaccination is because HCPs believe that the safety and efficacy of the vaccine, as well as the severity of illness, contribute to the acceptance of the vaccine [4,37,38,39].

### 4.4. Reasons

We found that a lack of time to become vaccinated was considered the main reason for hesitating to receive the flu vaccine. Other studies in the literature have not investigated the reasons related to a lack of time, although this problem has just been highlighted by the WHO in its program to contrast VH by which it proposes measures to contrast through an active call or reminder and an advisory service [3,5]. Equally important, according to the Italian experience, were the information campaigns [5]. For some causes, only logistical measures are needed, such as reducing costs or improving access by extending opening hours, or scheduling immunization appointments at strategic points to meet students’ needs [3]. For example, opening immunization centers in the afternoon could increase vaccination coverage, as could the establishment of a computerized immunization register to quickly track reluctant populations [5].

Consistent with other studies, we found that the role of adjuvant and long-term risks were less common reasons for VH [36,39]. The fears caused by these events still go back to the Wakefield study in 2004 [39], but also to the belief that vaccines may contain dangerous substances and that adverse events are not taken into account [36]. More recently, the belief of some people about the need to be vaccinated disappeared for low-risk diseases, which emerges as an important factor for VH, without considering all the protective mechanisms of vaccines [36]. This can be countered through discussion, information, and education [5].

### 4.5. Limitations

The limitations are those that are characteristic of cross-sectional studies. First, our results are based on anamnestic information from students. This may lead to underestimation or overestimation of a phenomenon. Second, we had a convenience sample because it consisted of a proportion of healthcare students; we chose to include only those most likely to relate to the patient in the study. Third, another limitation of the study could be the bias of the central tendency due to the face-to-face interview, but we are confident that the sample size can protect our results from this bias. Another limitation could be the influence of response bias on the results, but we used an anonymous questionnaire.

## 5. Conclusions

Improving HCPs’ knowledge and ensuring their access to information is critical to supporting HCPs’ role as “trusted ambassadors” who can promote vaccine acceptance, answer questions, or have difficult conversations with those who do not want to be vaccinated. There is a need to promote various approaches, most of which are suggested in the WHO report, to help HCPs and future HCPs better understand this issue that is so important to our public health and well-being. In summary, it is essential to pay attention to the vaccine/vaccination issue, not only in terms of the possible determinants of VH and fake news, but also in terms of knowledge of the entire vaccine production cycle, from initial testing in the laboratory to human application and the continuous process of reporting adverse effects with the aim of controlling and limiting the phenomenon of VH. Given the differences we have highlighted between men and women that may indicate gender differences, future studies should be designed to explore these in more detail.

## Figures and Tables

**Figure 1 ijerph-20-00533-f001:**
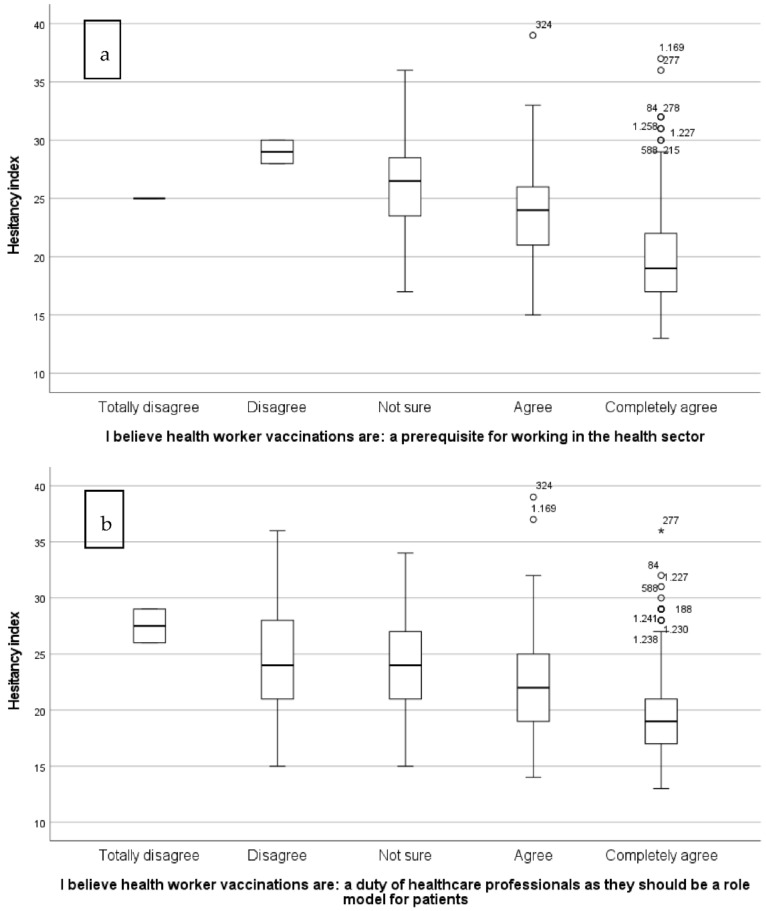
VH index distribution by the opinion of healthcare professionals toward vaccinations.

**Table 1 ijerph-20-00533-t001:** Socio-demographic characteristics of the respondents (*n* = 1275) by vaccine hesitancy level.

Variables	Total	Vaccine Hesitancy			*p*-Value (Cramer’s V)
		≤18	19–22	≥23	
		*n* (%)	*n* (%)	*n* (%)	
**Sex**					
Female	684 (53.7)	238 (54.7)	234 (53.4)	212 (52.9)	0.860 ^a^
Male	590 (46.3)	197 (45.3)	204 (46.6)	189 (47.1)	
**Median age** (IQR)	21 (19–22)	21 (20–23)	21 (20–22)	20 (19–22)	<0.001 ^b^
**Degree**					
Physiotherapy	84 (6.6)	22 (5.1)	35 (8.0)	27 (6.7)	<0.001 ^a^ (0.239) ^c^
Nursing	273 (21.4)	56 (12.9)	106 (24.2)	111 (27.7)	
Medicine	647 (50.8)	306 (70.3)	214 (48.9)	127 (31.7)	
Obstetrics	41 (3.2)	6 (1.4)	17(3.9)	18 (4.5)	
Physical education	199 (15.6)	34 (7.8)	59 (13.5)	106 (26.4)	
Biomedical laboratory technicians	30 (2.4)	11 (2.5)	7 (1.6)	12 (3.0)	

^a^ Chi-square test. ^b^ Kruskal–Wallis’ test. ^c^ Cramer’s V.

**Table 2 ijerph-20-00533-t002:** Do you know which of the following vaccinations are recommended for healthcare professionals?

Vaccine	Answers	VH			*p*-Value (Cramer’s V)
		<18 *n* (%)	18–22 *n* (%)	>23 *n* (%)	
**Seasonal flu**	Yes	>257 (59.1)	238 (54.3)	184 (45.9)	0.003 (0.08)
	I don’t know	60 (27.1)	59 (32.2)	63 (38.4)	
	No	118 (13.8)	141 (13.5)	154 (15.7)	
**Chickenpox**	Yes	321 (73.8)	299 (68.3)	222 (55.4)	<0.001 (0.12)
	I don’t know	89 (20.5)	107 (24.4)	146 (36.4)	
	No	25 (5.7)	32 (7.3)	33 (8.2)	
**MMR**	Yes	384 (88.3)	365 (83.3)	288 (71.8)	<0.001 (0.13)
	I don’t know	49 (11.3)	65 (14.8)	101 (25.2)	
	No	2 (0.5)	8 (1.8)	12 (3)	
**Hep B**	Yes	381 (87.6)	370 (84.5)	298 (74.3)	<0.001 (0.11)
	I don’t know	50 (11.5)	67 (15.3)	99 (24.7)	
	No	4 (0.9)	1 (0.2)	4 (1)	
**Hep A**	Yes	221 (50.8)	223 (50.9)	219 (54.6)	0.002 (0.08)
	I don’t know	143 (32.9)	172 (39.3)	150 (37.4)	
	No	71 (16.3)	43 (9.8)	32 (8)	
**Tdap**	Yes	364 (83.7)	361 (82.4)	279 (64.6)	<0.001 (0.11)
	I don’t know	63 (14.5)	71 (16.2)	116 (28.9)	
	No	8 (1.8)	6 (1.4)	6 (1.5)	
**Pneumococcal**	Yes	262 (60.2)	239 (54.6)	185 (46.1)	<0.001(0.09)
	I don’t know	153 (35.2)	180 (41.1)	200 (49.9)	
	No	20 (4.6)	19 (4.3)	16 (4.0)	
**Meningococcal**	Yes	330 (75.9)	297 (67.8)	223 (55.6)	<0.001 (0.13)
	I don’t know	97 (22.3)	130 (29.7)	169 (42.1)	
	No	8 (1.8)	11 (2.5)	9 (2.2)	
**BCG**	Yes	286 (65.7)	271 (61.9)	208 (51.9)	<0.001 (0.12)
	I don’t know	118 (27.1)	149 (34.0)	184 (45.9)	
	No	31 (7.1)	18 (4.1)	9 (2.2)	

Notes: MMR: Measles, Mumps, and Rubella. Hep B: the Hepatitis B vaccine. Hep A: the Hepatitis A vaccine. Tdap: Tetanus, Diphtheria, and Pertussis. BCG: Bacillus Calmette–Guérin.

**Table 3 ijerph-20-00533-t003:** Knowledge distribution by sex.

Vaccine	Total Answers “Yes” *n*(%)	Males Answer “Yes” *n*(%)	Females Answer “Yes” *n*(%)	*p*-Value (Cramer’s V)
**Seasonalflu**	679 (53.3)	290 (49.2)	389 (56.9)	0.016 (0.081)
**Chickenpox**	842 (66.1)	353 (59.8)	489 (71.5)	<0.001 (0.125)
**MMR**	1037 (81.4)	444 (75.3)	593 (86.7)	<0.001 (0.148)
**Hep B**	1049 (82.3)	451 (76.4)	598 (87.4)	<0.001 (0.144)
**Hep A**	663 (52.0)	288 (48.8)	375 (54.8)	0.066 (0.065)
**Tdap**	1004 (78.8)	442 (74.9)	562 (82.2)	0.007 (0.088)
**Pneumococcal**	686 (53.8)	290 (49.2)	396 (57.9)	0.004 (0.093)
**Meningococcal**	850 (66.7)	359 (60.8)	491 (71.8)	<0.001 (0.117)
**BCG**	765 (60.0)	321 (54.4)	444 (64.9)	<0.001 (0.109)

Notes: MMR: Measles, Mumps, and Rubella. Hep B: the Hepatitis B vaccine. Hep A: the Hepatitis A vaccine. Tdap: Tetanus, Diphtheria, and Pertussis. BCG: Bacillus Calmette–Guérin.

## Data Availability

Not applicable.

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
