# Peer review of "Healthcare University Courses Fail to Improve Opinions, Knowledge, and Attitudes toward Vaccines among Healthcare Students: A Southern Italy Cross-Sectional Study"

_ijerph, 2022, doi:10.3390/ijerph20010533_

Round 1
Reviewer 1 Report
The paper is written according to the needs on a global basis. From my point of view there is a lack of literature review, without any structure. Do you have any variable to describe in literature review under the specific title after the introduction part. Also, the sentences which are "the water is wet" doesn't contribute to excellence of the research. Cronbach alpha should not be prepared in abstract. Conclusion from the abstract doesn't describe any aspect of the results. Discussion has a goo structure. Maybe literature review should be prepared according to discussion part.
Author Response
Reviewer 1
The paper is written according to the needs on a global basis. From my point of view there is a lack of literature review, without any structure. Do you have any variable to describe in literature review under the specific title after the introduction part. Also, the sentences which are "the water is wet" doesn't contribute to excellence of the research.
The reviewer makes a valid point; therefore we have revised the introduction with some elements of integration relating to the role of training in the university context, also underlining its greater effectiveness in this subjects category.
Regarding the title, our aim was to highlight the lackness in the university training of future health professionals. The results obtained represent a starting point for the implementation of more effective training.
Cronbach’s alpha should not be prepared in abstract.
The reviewer makes a valid point; therefore we eliminated Cronbach’s alpha from the abstract.
Conclusion from the abstract doesn't describe any aspect of the results.
The reviewer makes a valid point; therefore, we have reviewed the abstract conclusion.
Discussion has a goo structure. Maybe literature review should be prepared according to discussion part.
We have prepared the introduction in accordance with the discussion by reporting the literature references relating to knowledge, opinion and attitudes with respect to vaccinations/vaccines, however we did not deem it appropriate to divide it into sub-paragraphs

Reviewer 2 Report
Thank you for the opportunity to read and review your research contribution. I find the topic very interesting and your research relevant. However, there are a number of aspects that should be taken into account:
1. The introduction is well planned, but it is necessary to delve deeper into the background in the university environment. Only briefly mentions the academic context.
2. In the methodology, in participants, more data is needed on the calculation of the sample size, why select a wide range of careers? Do these students receive vaccine training in their career? The questionnaire he uses has been applied to professionals, but not to students. Please provide data on the validity of the instrument in your sample. Please include information about the procedure and ethical aspects of the research. In general, the methodology should be reviewed and expanded.
3. Too many irrelevant figures have been included.
4. In the discussion, I would recommend that you start by remembering the objective of the study.
5. The figure of the questionnaire appears above the text.
Thank you for the opportunity to read and review your research contribution. I find the topic very interesting and your research relevant. However, there are a number of aspects that should be taken into account:
1. The introduction is well planned, but it is necessary to delve deeper into the background in the university environment. Only briefly mentions the academic context.
2. In the methodology, in participants, more data is needed on the calculation of the sample size, why select a wide range of careers? Do these students receive vaccine training in their career? The questionnaire he uses has been applied to professionals, but not to students. Please provide data on the validity of the instrument in your sample. Please include information about the procedure and ethical aspects of the research. In general, the methodology should be reviewed and expanded.
3. Too many irrelevant figures have been included.
4. In the discussion, I would recommend that you start by remembering the objective of the study.
5. The figure of the questionnaire appears above the text.
Author Response
Reviewer 2
- The introduction is well planned, but it is necessary to delve deeper into the background in the university environment. Only briefly mentions the academic context.
The reviewer makes a valid point; therefore we have revised the introduction with some elements of integration relating to the role of training in the university context, also underlining its greater effectiveness in this category of subjects.
- In the methodology, in participants, more data is needed on the calculation of the sample size, why select a wide range of careers?
We have decided to propose all students of the health professions participation in the study, therefore we did not consider it useful to calculate the sample size.
We decided to include all careers involved in healthcare in the study because we believe that each of them can play an important role in tackling vaccine hesitancy.
Do these students receive vaccine training in their career?
To date these students have not received any training, for this reason we have carried out this survey which has provided us with useful elements for organizing a future training.
The questionnaire he uses has been applied to professionals, but not to students. Please provide data on the validity of the instrument in your sample.
Validation of the questionnaire was carried out verifying the understanding of the questions on a small pilot sample of future health professionals (pre-test method).
We think it is possible to state that the estimated vaccine hesitancy accuracy was not influenced by the questions used because our vaccine hesitancy index included only the answers to questions relating to personal opinions on the topic (Question n. 1 of Appendix 1).
As regards the specific questions on recommended vaccinations (Question n. 4 of Appendix 1) we think that students should know this information because the Italian law provides some obligatory and recommended vaccinations for them as well.
Furthermore, the questions related to potential side effects (Question n. 8 of Appendix 1) were intended to investigate their knowledge at the time of the study and the possible influence of fake news.
Please include information about the procedure and ethical aspects of the research.
We have written that “All procedures were performed according to the ethical standards of the 1975 Helsinki Declaration, as revised in 2013. Ethical approval was given by the Ethics Committee of the University Hospital “Policlinico G. Rodolico – S.Marco” of Catania. Written informed consent was obtained from all participants”.
In general, the methodology should be reviewed and expanded.
The reviewer makes a valid point; therefore, we have reviewed and expanded the methodology.
- Too many irrelevant figures have been included.
The reviewer makes a valid point; therefore, we have renamed Figures 2a-f and Figures 3a-c as Figures S1a-f and S2 a-c, respectively
- In the discussion, I would recommend that you start by remembering the objective of the study.
The reviewer makes a valid point; therefore, we have included the objective of the study at the beginning of discussion.
- The figure of the questionnaire appears above the text.
Unfortunately we cannot fix the figure because it is not superimposed on the writing in the word file. We suppose it is a problem due to the conversion into pdf automatically performed by the journal site during the submission of the manuscript

Reviewer 3 Report
- Courses Fail or course fails
- Add study setting to the title
- English needs to be improved, it is hard to follow
- Add number of students to the abstract
- Line 107 . Unnecessary
- Improve quality of figure 1
- Please add literature from last few years as this is emerging topic
Author Response
Reviewer 3
- Courses Fail or course fails
Done
- Add study setting to the title
Done
- English needs to be improved, it is hard to follow
Done
- Add number of students to the abstract
Done
- Line 107 . Unnecessary
Done
- Improve quality of figure 1
Done
- Please add literature from last few years as this is emerging topic
We have added some references in the introduction highlighting the impact of the pandemic on vaccination hesitancy both in the general population and on some health professionals.

Round 2
Reviewer 1 Report
The authors improved the introduction part, while explaining the literature review in addition with the comments. Also, the suggestions on the abstract are improved.
Author Response
Reviewer 1
The authors improved the introduction part, while explaining the literature review in addition with the comments. Also, the suggestions on the abstract are improved.
We thank the reviewer for the positive comment.

Reviewer 2 Report
The figure of the questionnaire appears above the text and is not very relevant, it should be deleted.
Author Response
Reviewer 2
The figure of the questionnaire appears above the text and is not very relevant, it should be deleted.
Done
